# Comparison of devices used to measure blood pressure, grip strength and lung function: A randomised cross-over study

Carli Lessof[1], Rachel Cooper[2,3], Andrew Wong[4], Rebecca Bendayan[5,6], Rishi Caleyachetty[7,8], Hayley Cheshire[9], Theodore Cosco[10], Ahmed Elhakeem[11], Anna L. Hansell[12], Aradhna Kaushal[13], Diana Kuh[4], David Martin[1], Cosetta Minelli[14], Stella Muthuri[4], Maria Popham[4], Seif O. Shaheen[15], Patrick Sturgis[16], Rebecca Hardy[17,18]*

1 National Centre for Research Methods, University of Southampton, Southampton, United Kingdom, 2 Faculty of Medical Sciences, Translational and Clinical Research Institute, AGE Research Group, Newcastle University, Newcastle upon Tyne, United Kingdom, 3 NIHR Newcastle Biomedical Research Centre, Newcastle University and Newcastle upon Tyne Hospitals NHS Foundation Trust, Newcastle upon Tyne, United Kingdom, 4 MRC Unit for Lifelong Health and Ageing at UCL, London, United Kingdom, 5 Department of Biostatistics and Health Informatics of the Institute of Psychiatry, Psychology and Neuroscience, King's College London, London, United Kingdom, 6 NIHR Biomedical Research Centre at South London and Maudsley, NHS Foundation Trust and King's College London, London, United Kingdom, 7 Oxford University Hospitals NHS Foundation Trust, Oxford, United Kingdom, 8 Warwick Medical School, University of Warwick, Warwick, United Kingdom, 9 Hayley Cheshire Research, Bournemouth, United Kingdom, 10 Department of Gerontology, Simon Fraser University, Vancouver, Canada and Oxford Institute of Population Ageing, University of Oxford, Oxford, United Kingdom, 11 MRC Integrative Epidemiology Unit at the University of Bristol, Bristol, United Kingdom, 12 Centre for Environmental Health and Sustainability, University of Leicester, United Kingdom, 13 Research Department of Behavioural Science and Health, UCL, London, United Kingdom, 14 National Heart & Lung Institute, Imperial College London, United Kingdom, 15 Institute of Population Health Sciences, Barts and The London School of Medicine and Dentistry, Queen Mary University of London, London, United Kingdom, 16 Department of Methodology, London School of Economics, United Kingdom, 17 Social Research Institute, UCL, London, United Kingdom, 18 School of Sport, Exercise and Health Sciences, Loughborough University, Loughborough, United Kingdom

* R.J.Hardy@lboro.ac.uk

**Data Availability Statement:** All relevant data are publicly available from the ReShare repository (https://dx.doi.org/10.5255/UKDA-SN-856306).

## Abstract

### Background

Blood pressure, grip strength and lung function are frequently assessed in longitudinal population studies, but the measurement devices used differ between studies and within studies over time. We aimed to compare measurements ascertained from different commonly used devices.

### Methods

We used a randomised cross-over study. Participants were 118 men and women aged 45–74 years whose blood pressure, grip strength and lung function were assessed using two sphygmomanometers (Omron 705-CP and Omron HEM-907), four handheld dynamometers (Jamar Hydraulic, Jamar Plus+ Digital, Nottingham Electronic and Smedley) and two spirometers (Micro Medical Plus turbine and ndd Easy on-PC ultrasonic flow-sensor) with multiple measurements taken on each device. Mean differences between pairs of devices

**Funding:** The project was funded by CLOSER (https://closer.ac.uk/), whose mission is to maximise the use, value and impact of longitudinal studies. CLOSER is funded by the UK Economic and Social Research Council (grant reference: ES/K000357/1). CL was funded by an ESRC Doctoral Training Programme grant at the University of Southampton (ES/J500161/1). The UK Medical Research Council funded AE, AK, AW, DH, MP (MC_UU_12019/1), RB, RH, RCa (MC_UU_12019/2) and RCo, SM, TC (MC_UU_12019/4) when undertaking this work. RB is funded in part by the King's College London UK Medical Research Council Skills Development Fellowship programme (MR/R016372/1) and by the National Institute for Health Research (NIHR) Biomedical Research Centre (IS-BRC-1215-20018) at South London and Maudsley NHS Foundation Trust and King's College London. The funders had no role in study design, data collection and analysis, decision to publish, or preparation of the manuscript.

**Competing interests:** The authors have declared that no competing interests exist.

were estimated along with limits of agreement from Bland-Altman plots. Sensitivity analyses were carried out using alternative exclusion criteria and summary measures, and using multilevel models to estimate mean differences.

## Results

The mean difference between sphygmomanometers was 3.9mmHg for systolic blood pressure (95% Confidence Interval (CI):2.5,5.2) and 1.4mmHg for diastolic blood pressure (95% CI:0.3,2.4), with the Omron HEM-907 measuring higher. For maximum grip strength, the mean difference when either one of the electronic dynamometers was compared with either the hydraulic or spring-gauge device was 4-5kg, with the electronic devices measuring higher. The differences were small when comparing the two electronic devices (difference = 0.3kg, 95% CI:-0.9,1.4), and when comparing the hydraulic and spring-gauge devices (difference = 0.2kg, 95% CI:-0.8,1.3). In all cases limits of agreement were wide. The mean difference in $FEV_1$ between spirometers was close to zero (95% CI:-0.03,0.03), limits of agreement were reasonably narrow, but a difference of 0.47l was observed for FVC (95% CI:0.53,0.42), with the ndd Easy on-PC measuring higher.

## Conclusion

Our study highlights potentially important differences in measurement of key functions when different devices are used. These differences need to be considered when interpreting results from modelling intra-individual changes in function and when carrying out cross-study comparisons, and sensitivity analyses using correction factors may be helpful.

## Introduction

Blood pressure, grip strength and lung function are commonly assessed in longitudinal population studies. All three are non-invasive measures of physiological function that are practical for a nurse or interviewer to administer in a home or clinical setting using portable equipment. They avoid the subjectivity of self-reports of health, enable researchers and clinicians to track changes in health and function over the life course [1] and are important biomarkers of healthy ageing [2]. Their repeat assessment within longitudinal studies, and inclusion in many studies, facilitates comparisons over time and across ages and cohorts [3,4].

Although there have been a number of initiatives to encourage standardisation of these measures [5–7], different devices have been adopted by different studies for a variety of practical reasons [8,9]. Furthermore, the device used within a long-running longitudinal study will often need to change over time as obsolete or outdated models are replaced with devices that are more technologically advanced and improve or extend measurement, are less costly, more portable or easier to use. Because devices of this kind are only subject to moderate regulation [10,11], the measures obtained from different makes and models of device are unlikely to be equivalent. This has important implications for research which either compares findings across studies or considers change in function longitudinally. For example, in a study modelling age-related changes in blood pressure across the life course which used data from eight British longitudinal studies, switching from a manual sphygmomanometer to an automated device, without correction for the difference in measurement, resulted in a steeper increase in mean trajectory of systolic blood pressure [4]. Similarly, artefactual findings attributable to a

change in device have been observed in studies of lung function [12,13]. Indeed, concerns about potential differences in measures due to differences in spirometry devices have contributed to study investigators in the UK discouraging within- and cross-study analyses [14,15].

There are existing studies which have shown differences between devices used to measure blood pressure [16–20], grip strength [21–24] and lung function [6,12,13,25–27], but these have not yet compared all the devices commonly used in cohort and longitudinal population studies in the UK and many other countries. Further, these are only occasionally discussed in the context of both within- and between-study comparisons. To address this gap, a randomised cross-over trial was undertaken to compare measurements between devices used to assess blood pressure, grip strength and lung function commonly used in UK longitudinal population studies within the CLOSER consortium [28].

## Methods

### Study design and sample

For each of blood pressure, grip strength and lung function, a randomised cross-over study was carried out, so as to make within-person measurement comparisons. The study was conducted following established (CONSORT) guidelines [29]. The target sample, based on sample size calculations (S1 Appendix), was 120 men and women from the general population aged 45 to 74 years comprising 20 men and 20 women from each of three age groups (45–54, 55–64, 65–74). Participants were drawn from a list of individuals who had participated in a market research study, consented to be re-contacted for research purposes, and were living in London and the South East of England. An invitation letter and information sheet was sent and this was followed-up with a telephone recruitment process including assessment of health-related exclusion criteria (S1 Appendix). Eligible participants were then invited to attend a face-to-face assessment and each participant was measured on every machine (Table 1) at a single assessment visit.

All 90-minute face-to-face assessments took place in central London between October 2015 and January 2016 and were conducted by one of seven researchers who were trained and tested in all relevant protocols. All participants gave informed, written consent. The analytical dataset was pseudo anonymised with each participant given a study number so that individuals could not be identified. Ethical approval for data collection was given by University College London (UCL) (Ethics Project Number: 6338/001) and, for analysis, by the University of Southampton (Ethics Project Number: 18498). Participants received feedback on their results, advice to contact their General Practitioner if their blood pressure was elevated, and a gift voucher.

During the assessment, each participant was assessed in the sequence shown in Table 2. Blood pressure was measured consecutively on each device and the remaining measures were ordered to ensure that there was sufficient time between the four grip strength and two spirometry measurements to avoid participant fatigue. Multiple measurements were recorded on

**Table 1. Makes and models of devices included in study.**

| Measurement | | | | |
|---|---|---|---|---|
| **Sphygmomanometer (blood pressure)** | Omron 705-CP | Omron HEM-907 | | |
| **Hand-held dynamometer (grip strength)** | Jamar Hydraulic analog hand dynamometer | Jamar Plus+ digital hand dynamometer | Nottingham electronic handgrip dynamometer | Smedley spring-gauge dynamometer |
| **Spirometer (lung function)** | Micro Medical Micro Plus turbine spirometer | ndd Easy on-PC ultrasonic flow-sensor spirometer | | |

**Table 2. Flow chart of assessment.**

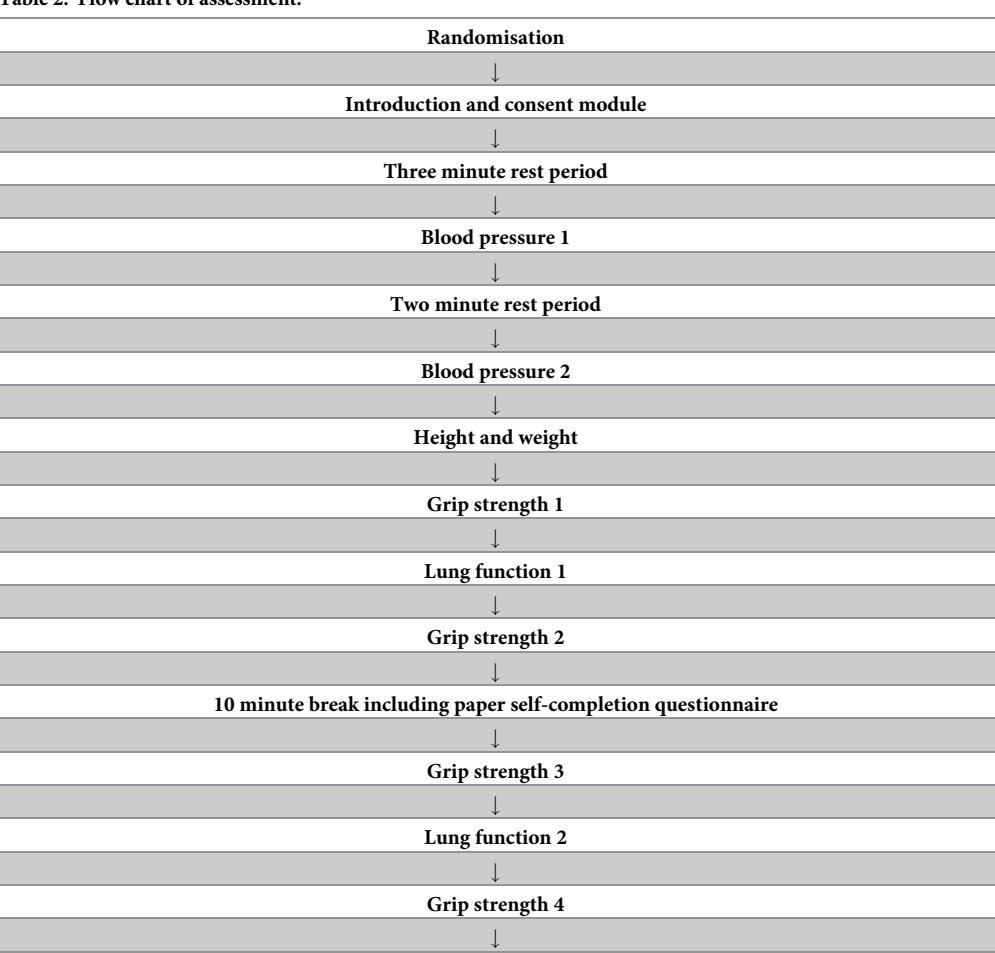

| |
|:---:|
| **Randomisation** |
| ↓ |
| **Introduction and consent module** |
| ↓ |
| **Three minute rest period** |
| ↓ |
| **Blood pressure 1** |
| ↓ |
| **Two minute rest period** |
| ↓ |
| **Blood pressure 2** |
| ↓ |
| **Height and weight** |
| ↓ |
| **Grip strength 1** |
| ↓ |
| **Lung function 1** |
| ↓ |
| **Grip strength 2** |
| ↓ |
| **10 minute break including paper self-completion questionnaire** |
| ↓ |
| **Grip strength 3** |
| ↓ |
| **Lung function 2** |
| ↓ |
| **Grip strength 4** |
| ↓ |
| **Copy of assessment data and gift voucher given to participant** |

each device as would be done in survey research. Height and weight were also measured and a short self-completion questionnaire was administered (S2 Appendix).

For each of the three measures, the order of devices was determined before fieldwork began, using computer-generated random numbers within each age-sex strata. Individuals were randomly allocated to one of two possible orders of blood pressure and lung function devices and to one of 24 possible orders of grip strength devices.

## Blood pressure, grip strength and lung function measurement

Standardised measurement protocols were used as follows. For blood pressure, the participant was asked to sit on a chair with legs uncrossed and their right arm resting comfortably, palm up, on a table, with the sphygmomanometers positioned so that they could not see the display. The participant was asked to expose their right arm, making sure that rolled up sleeves did not restrict circulation and that any watches or bracelets had been removed and, the sphygmomanometer cuff was then positioned over the brachial artery. After 3 minutes of quiet rest, 3 readings with a minute's rest between each reading were recorded using the first device. The device was then changed and after a further 2 minutes rest, 3 readings were taken using the second device. There was no talking until three readings on both devices had been completed.

Grip strength assessment was based on a published measurement protocol [30]. While seated in a chair with fixed arms, participants were asked to place their forearm on the arm of the chair in the mid-prone position (the thumb facing up) with their wrist just over the end of the arm of the chair in a neutral but slightly extended position. Adjustments were made to each dynamometer to accommodate different hand sizes according to the make and model of the device. On hearing the words "And Go", the participant was encouraged, through strong verbal instruction, to squeeze as hard as possible for a few seconds until told to stop. For each device, two measurements were carried out in each hand in the sequence Left-Right-Left-Right. The value on the display was recorded to the nearest 0.1kg for the Jamar Plus+ and Nottingham Electronic, to the nearest 0.5kg for the Smedley and to the nearest 1kg for the Jamar Hydraulic.

Lung function measurements adhered to the American Thoracic Society/European Respiratory Society (ATS/ERS) lung function protocol [6]. The procedure was explained and demonstrated, and the participant then had a practice blow without completely emptying their lungs. All measurements were carried out with the participant standing unless they felt unable to do so. During measurement, maximum effort was encouraged verbally. In addition, the ndd Easy on-PC was linked to a laptop which showed a cartoon of a child blowing up a balloon. This represents a real-time trace and as the participant is encouraged to exhale until the balloon pops this helps ensure a maximal FVC is achieved. After each trial the researcher recorded whether it satisfied the protocol, for example a trial was classified as not valid if the participant did not form a tight seal around the mouthpiece or coughed during the procedure, and in these instances, feedback was provided before the next attempt. Participants had up to five attempts to produce three valid measurements of lung function from each spirometer.

Readings for blood pressure, grip strength and lung function using the Micro Medical spirometer were data entered twice, independently, and compared to ensure accuracy. Lung function readings taken using the ndd Easy on-PC spirometer were downloaded directly from the laptop.

### Other measures

Height was measured using a portable Marsden Leicester stadiometer and weight using Tanita 352 scales according to standardised procedures, from which body mass index was calculated as weight (kg)/height (m)$^2$. Responses to the self-completion questionnaire provided additional information on: age at completing full-time education, self-rated health, smoking history, medication use and musculoskeletal, cardiovascular and respiratory conditions which might influence performance on the functional tests (S2 Appendix).

### Primary outcome measures

For the purposes of the main analyses, outcomes commonly used in epidemiological research were derived. The mean of the second and third readings of systolic blood pressure and diastolic blood pressure in millimeters of mercury (mmHg) were used. For grip strength, the maximum of the four readings in kilograms (kg) was used. For lung function, the maximum forced expiratory volume in 1 second (FEV$_1$) and forced vital capacity (FVC) in millilitres (ml) from the highest quality readings (quality A or B) were used. Quality grade A was when 3 or more acceptable tests were achieved with repeatability within 100 ml, and B when 3 acceptable tests were achieved with repeatability within 150 ml, as per ATS/ERS criteria [6].

### Statistical analyses

We described relevant characteristics by randomisation group for each measure. For each device we estimated the reliability using intraclass correlations (or Rho) and within-person

standard deviations using a variance-components model [31]. To investigate order effects we used two sample t-tests to compare the difference in mean values between groups with the measurements carried out in one sequence (device A followed by device B) compared with the opposite order (BA). For grip strength where 4 devices were tested, 6 pairwise comparisons were made, ignoring the exact placement of devices within the sequence.

We calculated the differences in measurement between pairs of devices then assessed the mean within-person differences between pairs of devices using paired t-tests. The assumption that the mean differences were normally distributed was checked by plotting histograms, and Bland-Altman plots (the difference between measures versus the average of the measures from the two devices for each individual) were used to assess whether the variation was dependent on the magnitude of the measurements [32,33]. The mean difference in values between the two devices, and the 95% limits of agreement, which give the range in which we would expect 95% of future differences in measurements between the two devices to lie, were plotted [33,34].

We also performed a series of sensitivity analyses to test the robustness of the results. We repeated analyses having: (i) excluded measurements where the devices were administered in the incorrect order (n = 2 for blood pressure, n = 5 for grip strength and n = 1 for lung function); (ii) removed extreme outliers identified using scatter plots (n = 1 for blood pressure and n = 2 for grip strength) and; (iii) used alternative outcome definitions commonly used in analyses. For blood pressure, we considered the mean of three readings [35] and the second reading only [36] and for grip strength, the mean of the four readings [37,38]. For lung function, we used the highest reading of $FEV_1$ and FVC drawn from all available readings irrespective of whether they adhered to the ATS/ERS quality criteria.

Finally, we used multilevel modelling, as an alternative statistical approach, to estimate the differences between devices, using all available readings rather than a summary measure, in order to account for variance between readings. The models treat the repeated readings as Level 1 and the individual as Level 2 to account for non-independence of measurements from the same person. Model 1 included device treated as a fixed effect. Model 2 also included covariates to account for the order in which the devices were administered and the position of the reading in the sequence (1 to 3 for blood pressure, 1 or 2 for the dominant and non-dominant hands for grip strength, and 1 to 5 for lung function). Model 3 was additionally adjusted for age, sex and, for blood pressure only, body mass index.

Data cleaning and management were carried out using Excel, IBM-SPSS Version 22 and STATA 14.0 and analyses were conducted using STATA 15.0.

## Results

During fieldwork, 118 assessments were completed, with 18–21 participants in each of the age-sex strata (S1 Table). Of the seven researchers, three carried out 20–30 assessments, two carried out 10–20 assessments and two carried out fewer than ten assessments.

The socio-demographic characteristics of the randomised groups were reasonably well balanced as were key aspects of cardiovascular, musculoskeletal and respiratory health (Tables 3 and 4). The reliability of every device was good. The intra-cluster correlations were lowest for blood pressure (0.89–0.94), due to the acknowledged within-person variation in this measure (S2 Table). The values for grip strength of dominant hand were above 0.95 for all devices except the Smedley dynamometer (0.92). Reliability was best for lung function (≥0.96), where within-person standard deviations were small. Reliability was slightly better when including only assessments adhering to the ATS/ERS quality criteria because two measures must be within 150ml of each other. There was no evidence of order effects for blood pressure or lung

**Table 3. Characteristics of the study population by first device used (N = 118).**

| First device | Blood pressure | | | | Grip Strength | | | | | | | | Lung function | | | |
|---|---|---|---|---|---|---|---|---|---|---|---|---|---|---|---|---|
| | Omron705-CP (n = 58[a]) | | Omron HEM-907 (n = 60) | | Jamar hydraulic (n = 30) | | Smedley (n = 28) | | Nottingham (n = 30) | | Jamar Plus+ (n = 30[a]) | | Micro Medical (n = 59) | | ndd Easy on-PC (n = 59[a]) | |
| | Mean | SD | Mean | SD | Mean | SD | Mean | SD | Mean | SD | Mean | SD | Mean | SD | Mean | SD |
| Age (years) | 59.4 | 8.2 | 59.8 | 7.8 | 58.5 | 8.2 | 59.8 | 7.4 | 59.0 | 9.0 | 61.2 | 7.4 | 59.8 | 7.8 | 59.4 | 8.3 |
| Weight (kg) | 76.9 | 21.1 | 77.3 | 16.7 | 73.8 | 16.1 | 82.1 | 22.2 | 77.3 | 17.6 | 75.5 | 19.4 | 77.3 | 16.5 | 76.9 | 21.2 |
| Height (cm) | 168.5 | 9.0 | 167.6 | 8.9 | 166.5 | 8.2 | 170.2 | 9.3 | 165.9 | 9.9 | 169.6 | 7.9 | 168.2 | 9.6 | 167.8 | 8.2 |
| BMI (kg/m$^2$) | 27.5 | 4.6 | 27.4 | 4.9 | 26.5 | 4.7 | 28.2 | 6.0 | 27.8 | 4.6 | 27.5 | 3.6 | 27.2 | 4.5 | 27.7 | 5.1 |
| | N | % | N | % | N | % | N | % | N | % | N | % | N | % | N | % |
| **Sex** | | | | | | | | | | | | | | | | |
| Men | 29 | 50 | 30 | 50 | 14 | 47 | 15 | 54 | 14 | 47 | 16 | 53 | 30 | 51 | 29 | 49 |
| Women | 29 | 50 | 30 | 50 | 16 | 53 | 13 | 46 | 16 | 53 | 14 | 47 | 29 | 49 | 30 | 51 |
| **Age (years) left full time education** | | | | | | | | | | | | | | | | |
| Under 16 | 20 | 34 | 19 | 32 | 6 | 20 | 8 | 29 | 11 | 37 | 14 | 47 | 24 | 41 | 15 | 25 |
| 17/18 | 7 | 12 | 10 | 17 | 5 | 17 | 6 | 21 | 3 | 10 | 3 | 10 | 5 | 8 | 12 | 20 |
| 19 + | 31 | 54 | 31 | 52 | 19 | 63 | 14 | 50 | 16 | 53 | 13 | 43 | 30 | 51 | 32 | 54 |
| **Self-reported health** | | | | | | | | | | | | | | | | |
| Excellent/ Very good | 32 | 55 | 32 | 53 | 17 | 57 | 15 | 54 | 18 | 60 | 14 | 47 | 32 | 54 | 32 | 55 |
| Good | 20 | 36 | 22 | 37 | 7 | 23 | 12 | 43 | 11 | 37 | 12 | 40 | 20 | 34 | 22 | 38 |
| Poor/Very poor | 6 | 10 | 6 | 10 | 6 | 20 | 1 | 4 | 1 | 3 | 4 | 13 | 7 | 12 | 4 | 7 |

[a] Sample size reduced by 1 for body mass index (BMI).

**Table 4. Cardiovascular, musculoskeletal and respiratory health status of the study population by first device used (N = 118).**

| First device | Omron 705-CP (n = 58) | | Omron HEM-907 (n = 60) | |
|---|---|---|---|---|
| | N | % | N | % |
| CV condition[a] | 4 | 7 | 9 | 15 |
| Hypertension | 18 | 31 | 19 | 32 |
| On blood pressure medication | 14 | 24 | 17 | 28 |

| First device | Jamar Hydraulic (n = 30) | | Smedley (n = 28) | | Nottingham (n = 30) | | Jamar Plus+ (n = 30) | |
|---|---|---|---|---|---|---|---|---|
| | N | % | N | % | N | % | N | % |
| Dominant hand–right | 29 | 97 | 25 | 89 | 27 | 90 | 27 | 90 |
| Arthritis | 6 | 20 | 5 | 18 | 4 | 13 | 5 | 17 |
| Some/ a lot of difficulty gripping | 5 | 17 | 8 | 29 | 6 | 20 | 5 | 17 |

| First device | Micro Medical (n = 59) | | ndd Easy on-PC (n = 59) | |
|---|---|---|---|---|
| | N | % | N | % |
| Respiratory condition[b] | 24 | 41 | 32 | 54 |
| On medication for condition | 4 | 7 | 2 | 3 |
| Current smoker | 13 | 22 | 8 | 14 |
| Ever smoker | 21 | 36 | 27 | 46 |

[a] Includes doctor diagnosed heart attack, angina and other heart condition

[b] Includes eczema, hay fever, asthma, COPD, bronchitis, emphysema and other respiratory problems.

**Table 5. Differences in mean and limits of agreement for each pair of devices used to measure blood pressure, grip strength and lung function.**

| Measures compared | | | | Limits of Agreement | |
|---|---|---|---|---|---|
| **Blood Pressure, mean of 2+3 (mmHg)** | N | Difference (95% CI) | p-value* | Lower | Upper |
| SBP, Omron HEM-907 –Omron 705-CP | 115 | 3.9 (2.5, 5.2) | <0.001 | -10.6 | 18.3 |
| DBP, Omron HEM-907 –Omron 705-CP | 115 | 1.4 (0.3, 2.4) | 0.01 | -9.8 | 12.5 |
| **Grip strength, max of 4 readings (kg)** | | | | | |
| Nottingham–Jamar Plus+ | 118 | 0.3 (-0.9, 1.4) | 0.6 | -12.1 | 12.7 |
| Jamar Hydraulic–Smedley | 118 | 0.2 (-0.8, 1.3) | 0.7 | -10.8 | 11.3 |
| Jamar Plus+–Jamar Hydraulic | 118 | 4.5 (3.9, 5.1) | <0.001 | -2.0 | 10.9 |
| Jamar Plus+–Smedley | 118 | 4.7 (3.7, 5.7) | <0.001 | -6.3 | 15.7 |
| Nottingham–Jamar Hydraulic | 118 | 4.7 (3.6, 5.9) | <0.001 | -7.9 | 17.3 |
| Nottingham–Smedley | 118 | 5.0 (3.5, 6.4) | <0.001 | -10.6 | 20.5 |
| **Lung function, maximum (litres), American Thoracic Society criteria** | | | | | |
| $FEV_1$, Micro Medical–ndd Easy on-PC | 74 | 0.00 (-0.03, 0.03) | 0.9 | -0.25 | 0.25 |
| FVC, Micro Medical–ndd Easy on-PC | 67 | -0.47 (-0.53, -0.42) | <0.001 | -0.92 | -0.03 |

* p-value from paired t-test.

function. For grip strength, there was evidence of an order effect for the comparison between the Nottingham Electronic and Smedley dynamometers (difference = -3.08kg (95% CI = -5.93, -0.23, p = 0.03) (S3 Table). Histograms show that for all three measures, the mean differences between devices were approximately normally distributed (S1 Fig).

## Blood pressure

Three participants were excluded from analyses due to missing readings leaving 115 for analysis. The mean difference in SBP between the two devices was 3.9mmHg (95% CI: 2.5, 5.2, p<0.001) and for DBP was 1.4mmHg (95% CI: 0.3, 2.4, p = 0.1), with the Omron HEM-907 measuring higher than the Omron 705-CP (Table 5). The Bland-Altman plots showed that as blood pressure increased, the difference between the two devices remained approximately constant (Figs 1 and 2). The limits of agreement were wide, being -10.6 to 18.3mmHg for SBP and -9.8 to 12.5mmHg for DBP.

## Grip strength

All 118 participants were included in the analyses. There was no evidence of a difference in mean maximum grip strength when comparing the two electronic dynamometers, the Nottingham Electronic and Jamar Plus+ (difference = 0.3kg (95% CI: -0.9, 1.4, p = 0.6), or when comparing the hydraulic and spring-gauge dynamometers, the Jamar Hydraulic and Smedley (difference = 0.2kg (95%CI:-0.8, 1.3, p = 0.7). However, there were mean differences in maximum grip strength of between 4 and 5kg when comparing either of the electronic dynamometers with either the hydraulic or spring-gauge dynamometer (Table 5). The limits of agreement varied depending on the pair of devices being compared, for example, these were narrower (-2.0 and 10.1 kg) when comparing the Jamar Plus+ and Jamar Hydraulic but very wide (-10.6 and 20.5 kg) when comparing the Nottingham Electronic and Smedley dynamometers. Even in cases where the mean difference was near zero, the limits of agreement indicated substantial differences in measurement between devices. The Bland-Altman plots (Figs 3–8) showed that for the comparisons of the Smedley dynamometer with all other devices, the difference increased at higher magnitudes of mean grip strength (Figs 4, 6 and 8).

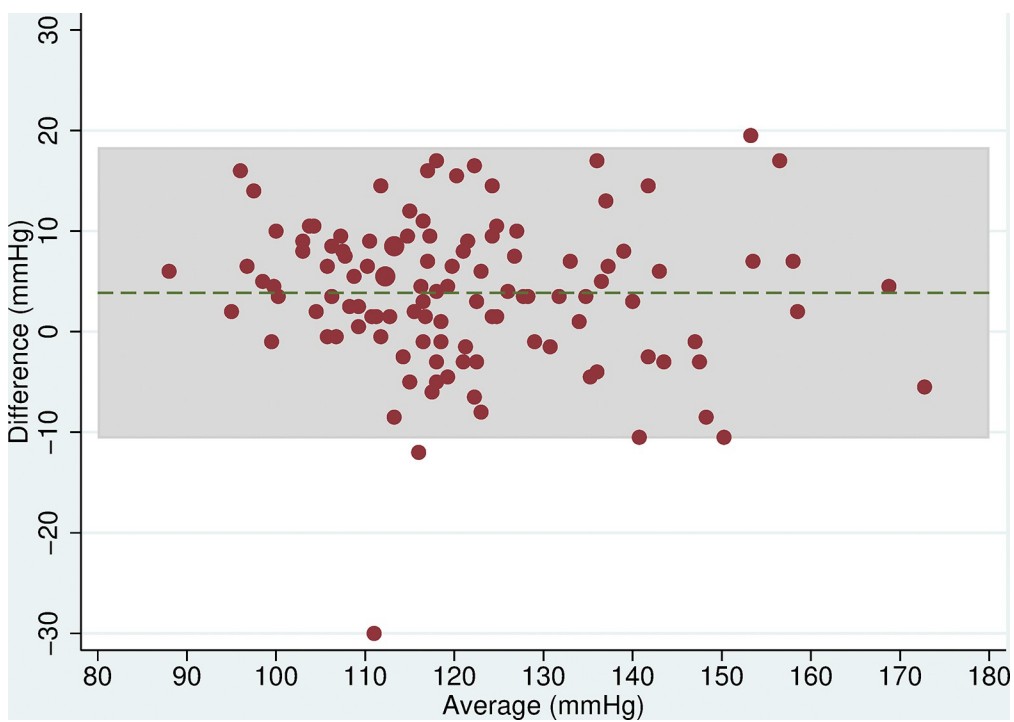

**Fig 1. Bland Altman plot for SBP.** Plot of the difference in mean SBP (mmHg) between the Omron 705-CP and Omron HEM-907 by the average SBP with 95% limits of agreement.

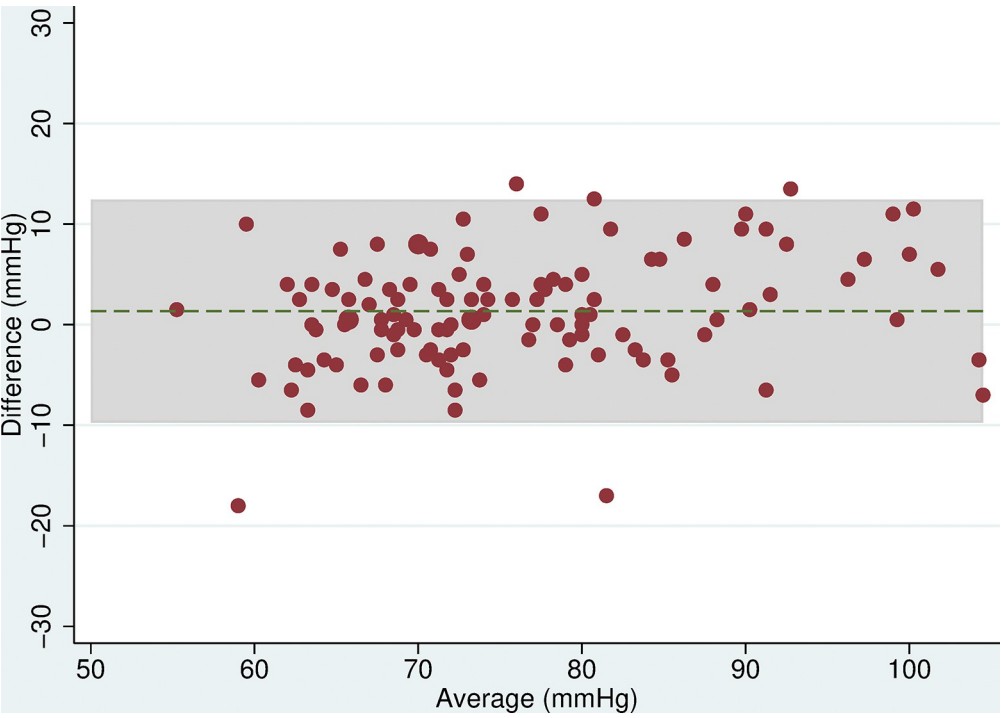

**Fig 2. Bland Altman plot for DBP.** Plot of the difference in mean DBP (mmHg) between the Omron 705-CP and Omron HEM-907 by the average DBP with 95% limits of agreement.

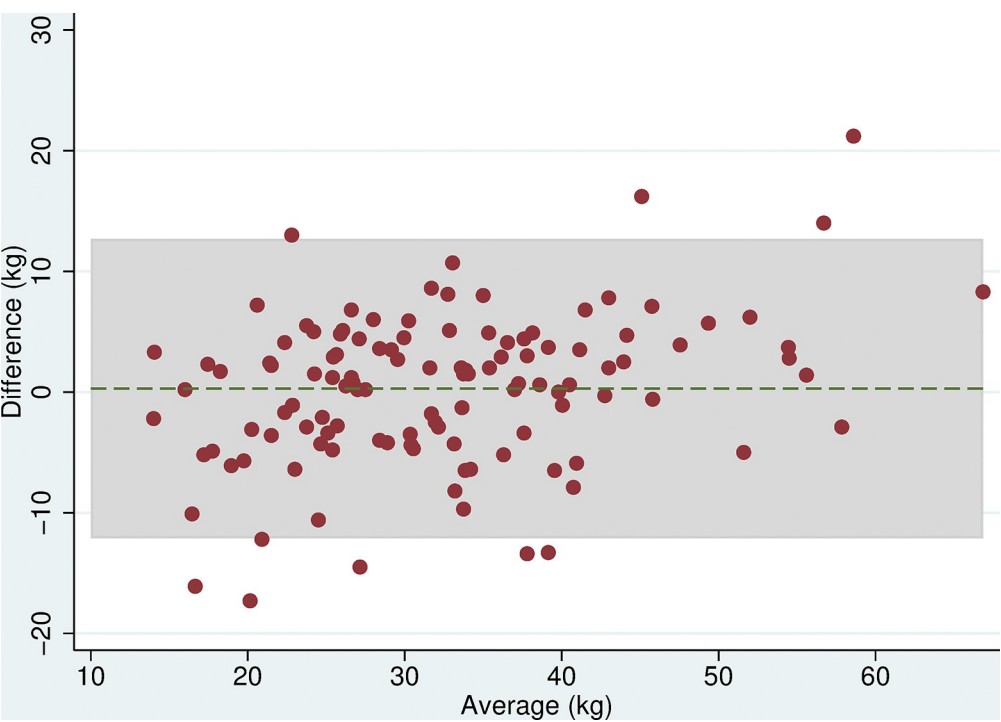

**Fig 3. Bland Altman plots of grip strength (Jamar Plus+–Nottingham).** Plot of the difference in maximum grip strength (kg) between devices by average maximum grip strength on both devices with 95% limits of agreement.

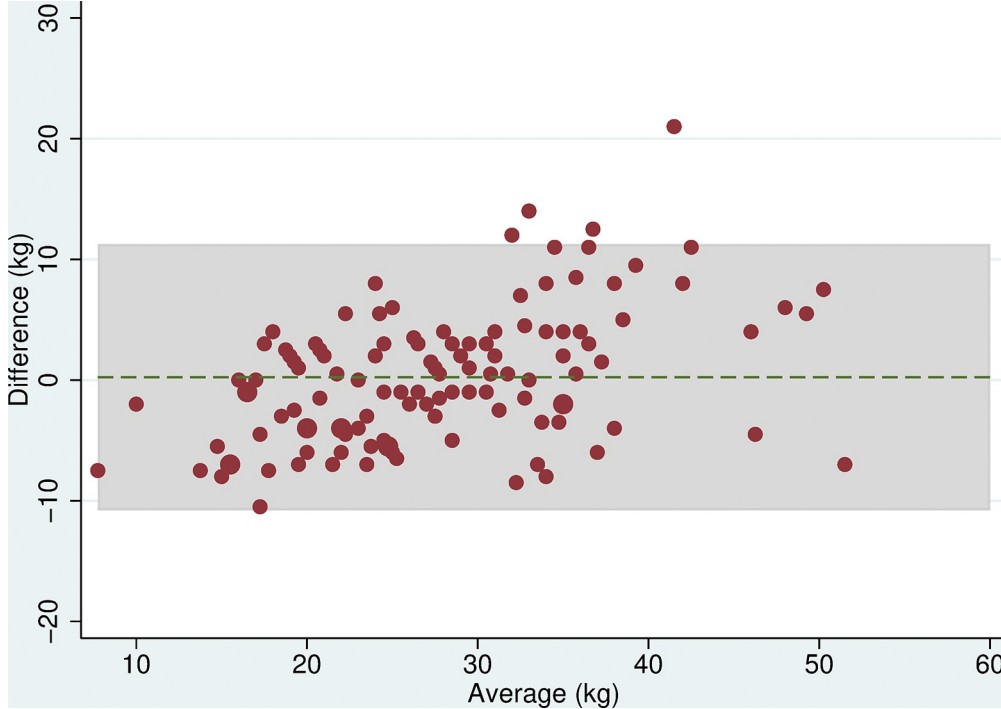

**Fig 4. Bland Altman plots of grip strength (Jamar Hydraulic–Smedley).** Plot of the difference in maximum grip strength (kg) between devices by average maximum grip strength on both devices with 95% limits of agreement.

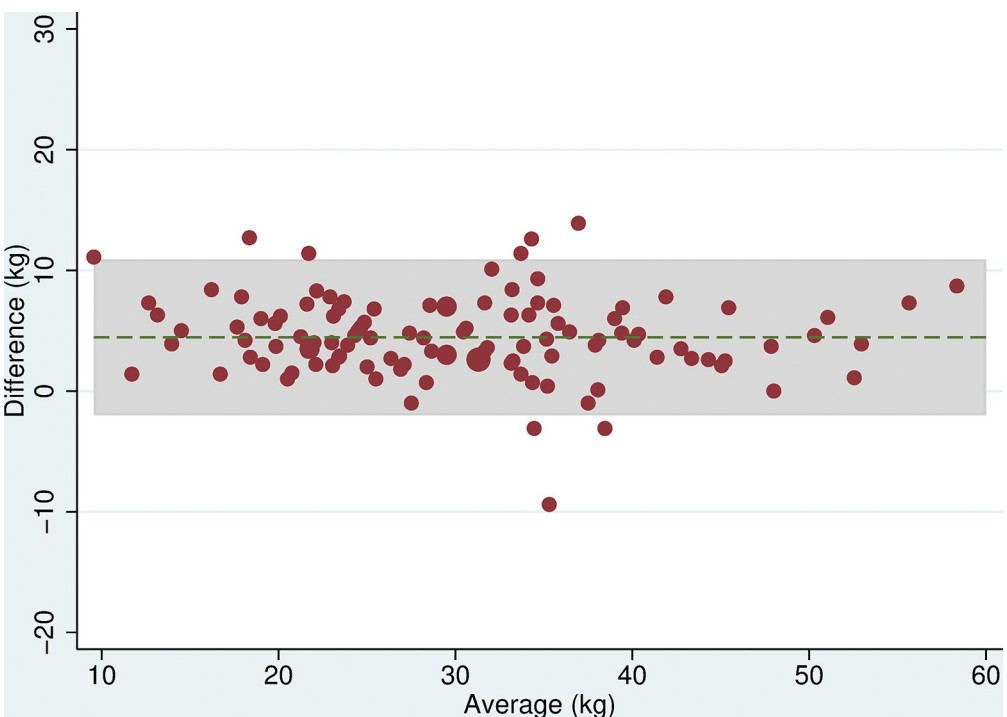

**Fig 5. Bland Altman plots of grip strength (Jamar Plus+–Jamar Hydraulic).** Plot of the difference in maximum grip strength (kg) between devices by average maximum grip strength on both devices with 95% limits of agreement.

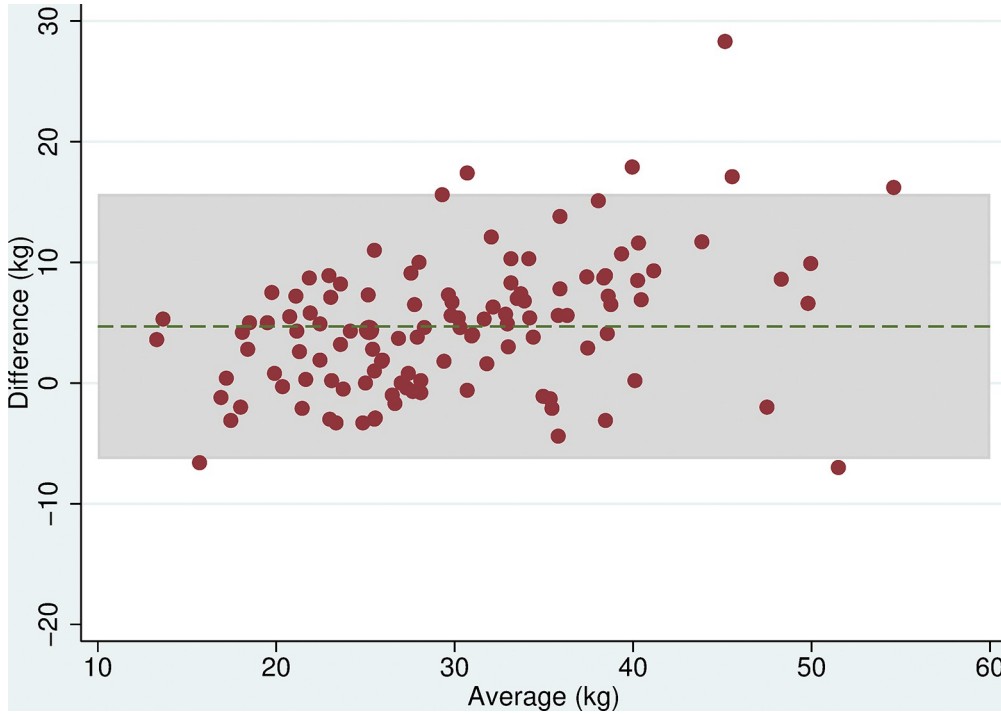

**Fig 6. Bland Altman plots of grip strength (Jamar Plus+–Smedley).** Plot of the difference in maximum grip strength (kg) between devices by average maximum grip strength on both devices with 95% limits of agreement.

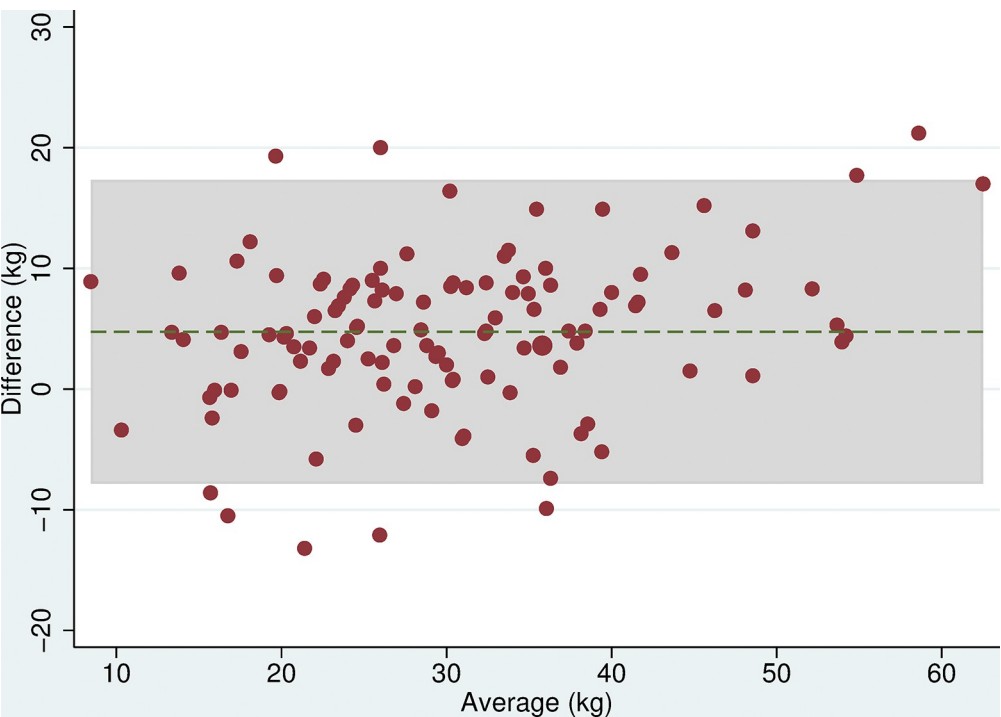

**Fig 7. Bland Altman plots of grip strength (Nottingham–Jamar Hydraulic).** Plot of the difference in maximum grip strength (kg) between devices by average maximum grip strength on both devices with 95% limits of agreement.

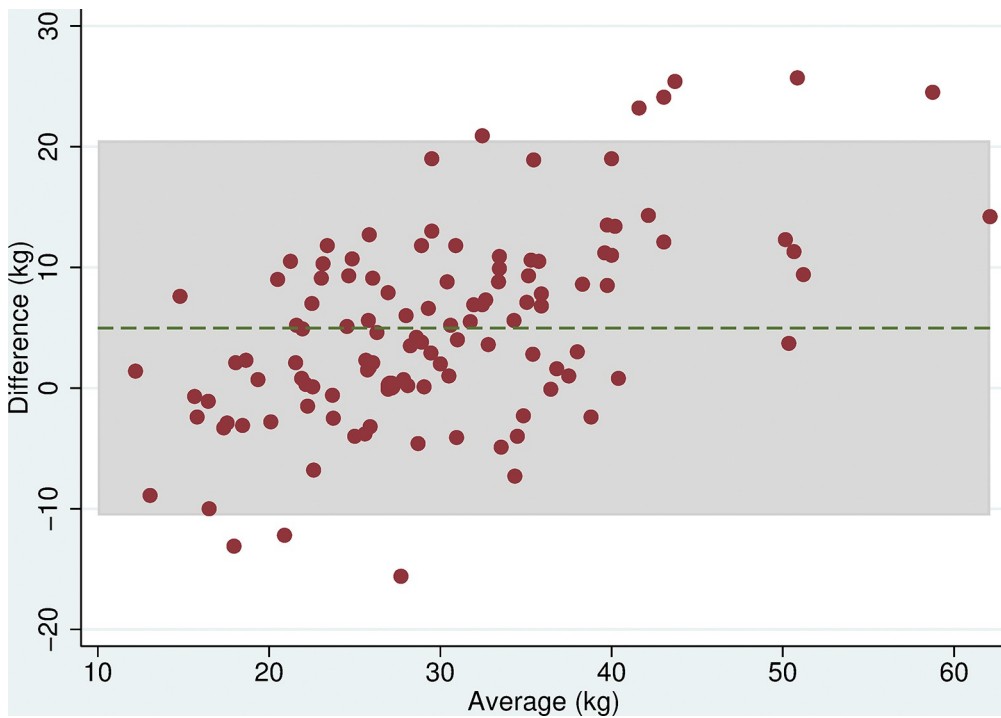

**Fig 8. Bland Altman plots of grip strength (Nottingham–Smedley).** Plot of the difference in maximum grip strength (kg) between devices by average maximum grip strength on both devices with 95% limits of agreement.

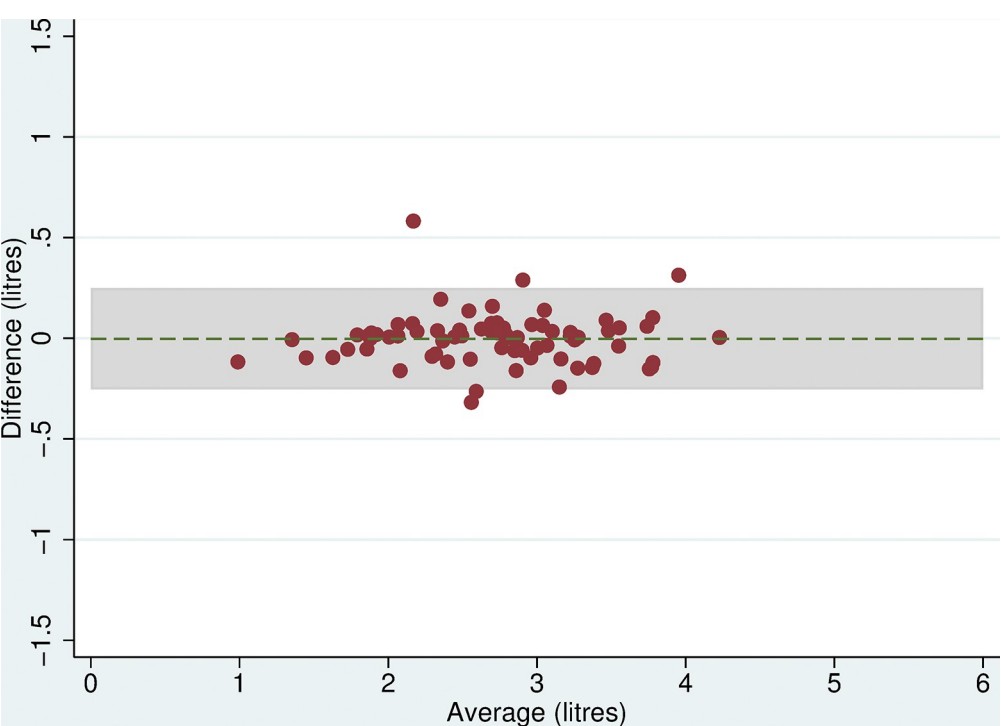

**Fig 9. Bland Altman plot for FEV₁.** Plot of the difference in mean maximum FEV1 between the Micro Medical and ndd Easy on-PC ultrasonic flow-sensor by average maximum $FEV_1$ on both devices with 95% limits of agreement.

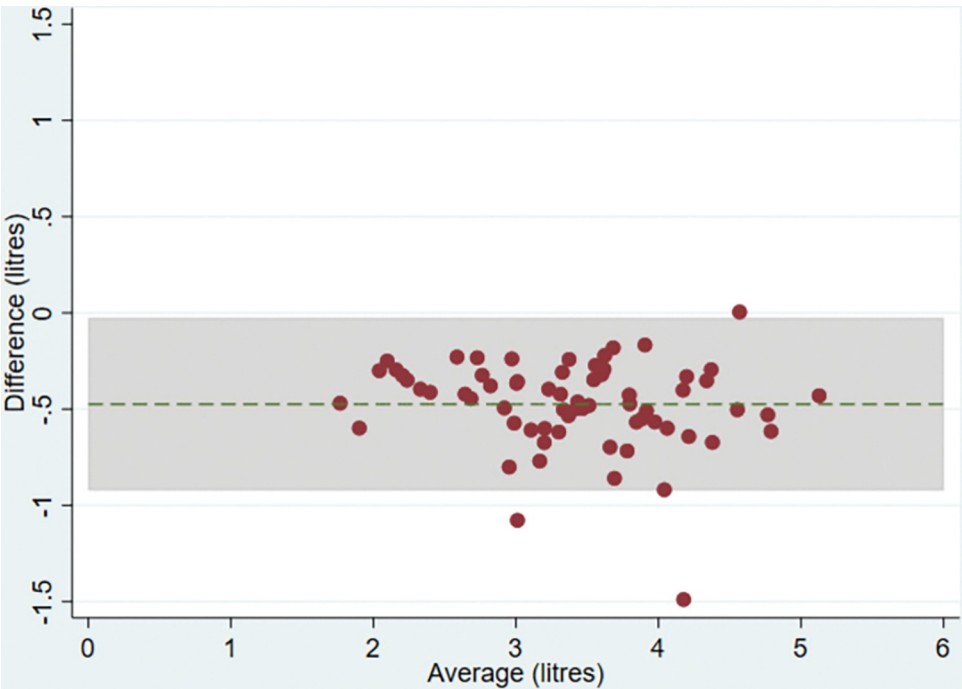

**Fig 10. Bland Altman plot for FVC.** Plot of the difference in mean maximum FVC between the Micro Medical and ndd Easy on-PC ultrasonic flow-sensor by average maximum FVC on both devices with 95% limits of agreement.

## Lung function

Twelve participants had missing lung function measures and just under a third (n = 32 for $FEV_1$ and n = 39 for FVC) of the remaining participants were excluded because there were no readings of a sufficiently high quality. There was no evidence of a difference in mean $FEV_1$ between devices (difference = 0.00 litres (95% CI:-0.03,0.03, p = 0.9)) but there was evidence of a difference in FVC (-0.47 litres (95% CI:-0.53,-0.42, p<0.001)) with the ndd Easy on-PC measuring higher than the Micro Medical (Table 5). The Bland-Altman plots suggested that for $FEV_1$, the difference between the two devices was approximately constant as measurements increased and close to zero (Fig 9) with reasonably narrow limits of agreement (-0.25 and 0.25 litres). The plot for FVC suggested that the difference between devices remained constant as values of FVC increased (Fig 10) but the limits of agreement were wider (-0.92 and -0.03).

## Sensitivity analyses

When we repeated the analyses having excluded measurements where the devices were administered in the incorrect order (n = 8), removed outliers (n = 3), included the lung function readings that did not meet ATS/ERS criteria (n = 32 for $FEV_1$ and n = 39 for FVC), and used alternative definitions of outcomes, there were only small changes in the estimated differences between devices such that the conclusions were unaltered (S4 Table). The only differences found were a small number of additional order effects (S5 Table), but these had no impact on the findings when order of device was controlled for through multilevel analysis. Indeed, when the data were reanalysed using multilevel models, the estimates of differences between devices showed only marginal changes, though the standard errors were reduced (S6–S8 Tables).

# Discussion

In a randomised cross-over study of 118 adults aged 45–74 years, we found evidence of differences in measurement of blood pressure, grip strength and lung function when assessed using different devices. For blood pressure, the newer Omron HEM-907 measured higher than the older Omron 705-CP with wide limits of agreement. For grip strength, the two electronic dynamometers recorded measurements on average 4-5kg higher than either the hydraulic or the spring-gauge dynamometer, but there were only small mean differences when comparing the two electronic dynamometers or the hydraulic and spring-gauge dynamometers. However, limits of agreement were wide for all comparisons. For lung function, the ndd Easy on-PC measures of FVC were an average of 0.47 litres higher than those for the Micro Medical, but there was no difference between measures of $FEV_1$ and the limits of agreement were reasonably narrow.

We are aware of only a few studies that have compared combinations of dynamometers previously. For example, King [21] compared the Jamar Hydraulic with the Jamar Plus+ dynamometer and, in contrast to our findings, reported that the electronic dynamometer had consistently lower readings than the hydraulic device and narrower. However, the study population was younger, with an average age of 32 years, comprising a convenience sample of 40 men and women and may have better function than our older sample which could influence comparability across machines. Another study reported a difference of 3.2kg (limits of agreement -6.3 to 12.6) when comparing the Smedley dynamometer and the Jamar Hydraulic dynamometer, which contrasts with our finding of a smaller mean difference (0.2kg) but wider limits of agreement (-10.8 to 11.3) [22]. However, this other study was carried out in an older, smaller sample of 55 participants aged 65–99 years recruited from a retirement home and social day care centre. Another study [23], found that the Smedley dynamometer measured lower than the Jamar+ Digital, similar to our study, although in this other study there

were other potentially important variations in measurement protocol–measures using the Smedley device were undertaken in a standing position and those using the Jamar device were undertaken seated. Our findings provide some reassurance that there is a lack of bias in measurement between specific device combinations (i.e. the Jamar Plus+ and Nottingham electronic; the Jamar Hydraulic and Smedley), although the limits of agreement suggest that the variation can still be substantial.

We have not identified a comparison of Micro Medical or other turbine spirometers with the ndd Easy on-PC spirometer. However, in a study of 35 volunteers, the Micro Medical turbine spirometer, used in our study, gave lower readings compared with the Vitalograph Micro pneumotachograph spirometer [13], for both $FEV_1$ (mean difference of 0.24l) and FVC (0.34l). Another study of 49 volunteers found that the handheld ndd Easy on-PC spirometer produced systematically lower values than a pneumotachograph spirometer (Masterscreen) [25], for both $FEV_1$ (mean difference of 0.24l) and for FVC (0.37l).

For lung function, the accuracy of measurement relies primarily on optimal coaching: maximally deep breath, a rapid blast and appropriate encouragement as well as a full seal around the mouthpiece and correct body posture [6]. The ndd Easy on-PC spirometer presents visualisation of the volume-time graph in real time, meaning that the participant can be encouraged to blow until the curve has reached a plateau, that is, when the true FVC has been achieved. In the absence of this visual display the forced manoeuvre may be terminated prematurely, and the FVC underestimated. We propose that this is the most likely explanation for the substantially higher FVC values obtained using the ndd Easy on-PC device than the Micro Medical device in our study, while there was no difference for $FEV_1$. For $FEV_1$ the mean difference between the 2 spirometers was zero and are, therefore, within the 150ml ATS/ERS criteria for replication of measurement. In addition, the limits of agreement did not exceed the 350ml criterion set in previous spirometry studies [27]. Whether using a group correction for FVC is valid, however, remains debateable as in the SAPALDIA study, a group correction from a quasi-experimental study was found not to be adequate, and an approach using spirometer-specific reference equations from longitudinal measurements to describe individualised corrections terms was preferred [12].

In considering the potential clinical significance of the differences between devices, we have referred to published normative or predicted values of blood pressure, grip strength and lung function [3,39,40]. Based on analysis of age-related differences in mean blood pressure in the Health Survey for England 2016, the mean differences in SBP and DBP between devices that we observed are equivalent to an age difference of approximately five years, although the possible non-linearity of change with age in diastolic blood pressure across the age range of interest [41] that comparison more difficult. Further, the within-person standard deviation for systolic blood pressure is larger than the mean difference between devices. For grip strength [3] the observed 4-5kg difference in grip strength is equivalent to an age difference of approximately 5 years among men and 10 years among women aged 65 years and above. For lung function, based on the National Health and Nutrition Examination Survey (NHANES) III data [42], predicted values for five-year age-groups (with male height of 175cm and female height of 160cm), show that a difference of 0.47l in FVC is equivalent to an age difference of around 15 years, between 45–75 years. Therefore, together with the wide limits of agreement and good measurement reliability for each device, the difference that we observed between devices are likely to have important practical implications for both grip strength and lung function. For example, the differences in dynamometers may result in discrepancies in clinical diagnoses which use cut-points when identifying an individual as sarcopenic [43]. Similarly, the difference in FVC, but not $FEV_1$, between machines will have implications for defining participants with COPD based on the ratio $FEV_1$/FVC.

Maintaining consistency in the make and model of device used in studies reduces the likelihood of measurement differences, but is not always realistic given that equipment becomes obsolete and new technology can improve measurement, for example through automation (as is the case with the Omron 907), the transition from analogue to digital (as is the case with the transition from the Jamar hydraulic to Jamar Plus+ devices) or the introduction of visual encouragement and specific feedback (as provided by the ndd Easy on-PC). An important implication of our findings is that it would be advisable for researchers, therefore, to include simple experiments to assess machine comparability when a new device is introduced into a study. Conducting external comparison studies, such as ours, would also help interpretation for both within-study and between-study comparisons. In addition, the differences between devices need to be considered in the context of reliability of measurements for each device being compared. Our analysis showed good reliability of measurements, particularly for the dynamometers and spirometers, suggesting the differences observed are important. The ATS/ ERS quality control for lung function ensures excellent reliability, but does result in exclusion of those who cannot meet the criteria.

A key strength of this study design was that it used the same standardised measurement protocols for all devices, which is important, as for all three functional measures, the type of device is only one of several factors which can affect measurements unless these other factors are kept constant as in our study. Blood pressure is affected by multiple factors [10] including the participant talking, actively listening, being exposed to cold, ingesting alcohol, having a distended bladder, recent smoking [44] and also to measurement protocols such as arm position and cuff size [45]. For grip strength, the values and precision of measurements have also been shown to be influence by a range of factors [30,37] including whether allowance is made for hand size and hand-dominance [46], dynamometer handle shape [47], position of the elbow [48] and wrist during testing [49], setting of the dynamometer [50,51], effort and encouragement, frequency of testing and time of day and training of the assessor [30,51]. The study also included a relatively large sample size, based on *a priori* sample size calculation, compared with other similar studies, and implemented a randomised design. While confidence in the results rests primarily on this randomised design [29], the fact that participants were drawn from a large database of members of the public, who had been involved in previous market research and consented to be re-contacted, suggests they may be more representative of the general population than the small-scale volunteer samples used in many previous studies. We also acknowledge the limitations of the study. The study findings cannot be generalised beyond the parameters of the research design; for example, results might differ for those outside the sampled age range (i.e., 45 to 74), and while the trial compared devices most commonly used in UK population-based studies, no comment can be made about device combinations which were not included [15]. While standardising the measurement protocols was an important aspect of the research design, it meant deviating from the protocol for the Smedley dynamometer (normally assessed standing rather than sitting) and so may limit the applicability of the findings for this device [30]. Furthermore, in the primary analyses of lung function, a number of participants were excluded due to missing or low-quality readings, particularly on the ndd Easy on-PC, thus reducing the sample size and power of these analyses. Nevertheless, sensitivity analyses using all available readings, irrespective of quality, suggested that this did not have a big impact on findings. Indeed, sensitivity analyses considering outliers, incorrectly ordered tests and alternative coding of measures, all showed that our results were robust. Assessor may be a source of variation in our study which we have not accounted for, although this variation was minimised by the consistent training and protocol, and is not likely to have had a substantial impact on differences between devices since this was a within-person comparison and the same researcher assessed the same person on all machines.

In conclusion, this randomised cross-over study showed measurement differences between devices commonly used to assess blood pressure, grip strength and lung function which researchers should be aware of when carrying out comparative research between studies and within studies over time.

## Supporting information

**S1 Table. Sample size by age group and sex.**
(DOCX)

**S2 Table. Reliability of the devices included in the study.**
(DOCX)

**S3 Table. Assessment of order effects for all measures.**
(DOCX)

**S4 Table. Sensitivity analysis for differences in mean and limits of agreement for all measures.**
(DOCX)

**S5 Table. Sensitivity analysis of order effects for all measures.**
(DOCX)

**S6 Table. Sensitivity analysis using multilevel models for blood pressure.**
(DOCX)

**S7 Table. Sensitivity analysis using multilevel models for grip strength.**
(DOCX)

**S8 Table. Sensitivity analysis using multilevel models for lung function.**
(DOCX)

**S1 Fig. Histograms of mean differences in SBP (mmHg), DBP (mmHg), grip strength (kg) and lung function ($FEV_1$ and FVC, litres) for all device combinations.**
(DOCX)

**S1 Appendix. Supplementary methods.**
(DOCX)

**S2 Appendix. Questionnaire.**
(DOCX)

## Acknowledgments

We thank the study participants, Kantar who provided access to the study sample, George Kyriakopoulos, now at BP (British Petroleum), for providing study advice, and Shaun Scholes, at UCL, for assistance with the analysis of Health Survey for England data referred to in the discussion.

## Author Contributions

**Conceptualization:** Carli Lessof, Rachel Cooper, Andrew Wong, Diana Kuh, Rebecca Hardy.

**Data curation:** Carli Lessof, Andrew Wong.

**Formal analysis:** Carli Lessof.

**Funding acquisition:** Andrew Wong, Diana Kuh, Rebecca Hardy.

**Investigation:** Carli Lessof, Andrew Wong, Rishi Caleyachetty, Theodore Cosco, Ahmed Elhakeem, Aradhna Kaushal, Stella Muthuri.

**Methodology:** Carli Lessof, Rachel Cooper, Andrew Wong, Rebecca Bendayan, Rishi Caleyachetty, Anna L. Hansell, Cosetta Minelli, Seif O. Shaheen, Patrick Sturgis, Rebecca Hardy.

**Project administration:** Andrew Wong, Hayley Cheshire, Maria Popham.

**Supervision:** David Martin, Patrick Sturgis, Rebecca Hardy.

**Validation:** Carli Lessof.

**Writing – original draft:** Carli Lessof.

**Writing – review & editing:** Rachel Cooper, Andrew Wong, Rebecca Bendayan, Rishi Caleyachetty, Hayley Cheshire, Theodore Cosco, Ahmed Elhakeem, Anna L. Hansell, Aradhna Kaushal, Diana Kuh, David Martin, Cosetta Minelli, Stella Muthuri, Maria Popham, Seif O. Shaheen, Patrick Sturgis, Rebecca Hardy.

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
